# Malleability and Physicochemical Properties of Industrial Sheet Margarine with Shea Olein after Interesterification

**DOI:** 10.3390/foods11223592

**Published:** 2022-11-11

**Authors:** Houbin Gao, Wei Gao, Xiaomin Yang, Yuanfa Liu, Zhouping Wang

**Affiliations:** 1State Key Laboratory of Food Science and Technology, School of Food Science and Technology, Jiangnan University, 1800 Lihu Road, Wuxi 214122, China; 2Wilmar (Shanghai) Biotechnology Research & Development Center Co., Ltd., Shanghai 200137, China; 3School of Materials Science and Engineering, Tianjin University of Technology, Tianjin 300384, China; 4School of Food Science and Technology, Jiangnan University, Wuxi 214122, China; 5Key Laboratory of Meat Processing of Sichuan, Chengdu University, Chengdu 610106, China

**Keywords:** industrial sheet margarine, malleability, polymorphism, shea olein, interesterification

## Abstract

The malleability of Industrial Sheet Margarine (ISM) is essential for the formation of consistent layered structures of pastry products. However, there is limited reporting on how to improve the malleability of ISM with zero trans fatty acids (TFA) at an industrial production scale. Therefore, herein, Shea Olein (SHOL), rich in stearic acid C18:0, was employed as a value-added formulation ingredient to replace palm olein (POL) in palm-based formula (palm stearin:palm kernel olein:palm olein, 50:15:35, *w*/*w*/*w*) and the chemical interesterification (CIE) fat as ISM material was performed to improve the application performance. The addition of SHOL improved the crystallization characteristics by increasing the β’ crystal content from 70.86% to 92.29% compared with a POL-added formula. The hardness of the two formulations after CIE decreased by 60% and 65%, respectively, compared with that before CIE due to the decrease of PPP and POP, and the increase of POS and PSS triacylglycerols. Melting profiles, polymorphism, and crystal structures systematically proved the bending and sheeting features and functional properties. The bending abilities and malleability of ISM with SHOL and CIE fats were significantly improved, resulting in a more conducive application performance. The study provided a practical approach to improving the malleability of ISM in industrial-level production.

## 1. Introduction

Sheet Margarine is widely applied in the baking industry to produce pastry products. Essentially, it is a roll-in margarine or Puff Pastry Margarine (PPM) with a regular square or rectangular shape. The most common weight and size format in industrial production is 1 kg per sheet with a length of 240–320 mm, a width of 240–280 mm, and a thickness of 10–15 mm. Generally, pastry is composed of multiple alternating fat/dough layers, which present an attractive flaky, layered structure after baking [1].

Sheet margarine should possess optimal plasticity and be moderately hard to lamellate without piercing the dough layers or breaking after laminating and folding during pastry preparation [2,3]. The fat belt should be kept continuous and not absorbed by dough during the sheeting process. Therefore, the quality of bakeries can be largely determined by the quality of sheet margarine [4,5]. The functionality (i.e., plasticity and malleability) of sheet margarine is intimately associated with its physicochemical properties such as solid fat content (SFC), polymorphism, and mechanical toughness of the crystal network [6].

Shea Olein (SHOL), considered a by-product of Shea stearin production, is often used as a downstream material for the production of biodiesel in the past years [7,8]. SHOL is one of the few vegetable oils which is naturally rich in stearic acid (C18:0 > 32%), typical fatty acid (FA) contents, and triacylglycerol (TAG) compositions are listed in Appendix A. According to previous studies, C18:0 can contribute to the acceleration of fat crystallization [9,10]. Therefore, SHOL is an underutilized and cost-saving material and holds great potential in the margarine production industry.

To obtain highly accepted laminated dough products, conventional industrial sheet margarine is usually manufactured with a formulation containing a high percentage of partially hydrogenated vegetable oils resulting in high levels of TFA in the products before 2018. Studies have indicated that a high intake of trans-fatty acids can increase serum cholesterol levels, leading to atherosclerosis, hypertension, and other cardiovascular diseases [11,12,13]. Currently, numerous research focuses on reducing TFA without losing the physical properties and functionality of sheet margarine. Oleogel-containing candelabra wax (CDW) was applied as a substitute for pastry fat in pastry products and acceptable dough and biscuit qualities were achieved [14]. Behic and Ilkem [15] compared the difference between oleogel and commercial roll-in shortening, achieving that cookies using oleogel to partly replace roll-in fat had a harder texture and higher spread rate. Sneh and Sanju Bala [16] used 20% olive oil to partially replace the hydrogenated fat of the original formulation. The aforementioned replacement altered the fatty acid composition of the product and increased its unsaturated fatty acid contents without compromising its sensory characteristics. Gao et al. [17] also improved the ductility of sheet margarine by strictly controlling the tempering process.

In addition, transesterification, as one of the three lipid modification technologies, is one of the means adopted to prepare zero TFA specialty bakery fat by repositioning the acyl arrangement of triglycerides while retaining the original composition of fatty acids to modify the properties of triglycerides without increasing the content of TFA. Oliveira and Rodrigues [18] conducted chemical interesterification (CIE) on the blends of Patawa oil and palm stearic acid to obtain the mixture containing 60–70% unsaturated fatty acids. Meanwhile, the mixture of 50% stearic acid and 50% Patawa oil after CIE had the desired properties such as suitable melting point, SFC, and plasticity. Adhikari [19] performed enzymatic interesterification (EIE) on the blend of pine kernel oil and palm oil using Lipozyme TL IM as a catalyst to prepare the zero-TFA product.

Lichtenstein and H. Alice [13] monitored the physical changes of shortening contained partially hydrogenated fat and concluded that the plasticity of the product was significantly improved through the tempering process. This conclusion can also be used to improve PPM malleability. However, partially hydrogenated fats which contain too high TFA would not be used in ISM owing to TFA issues. The physicochemical properties of CIE and Non-CIE palm-based sheet margarine was evaluated by Gao et al. [17] and they concluded that the CIE samples had better spread-ability compared with Non-CIE product, but post-hardening occurred for both formulations during storage led to bad ductility in shelf life.

Up to now, numerous research has also been carried out to meet the nutrition and functionality requirements of pastry fat. Although previous studies provided various solutions at the laboratory-scale level, these solutions still represented numerous defects regarding the perspective of functionality, industrialized production, actual commercialization such as cost, and materials source [20,21,22] and they have not been applied in actual industrial production to achieve higher nutrition and functionality [23,24,25,26]. Therefore, redesigning practical recipes that can be applied in mass production is crucial to the baking fat industry.

In this study, SHOL was introduced into the formulation to improve the application performance of the ISM compared to the usual palm oil ingredients. CIE was adopted to modify the base oil since it has been proven to be an effective and mature way to prepare zero trans low saturated fat. The differences (such as the composition of triglyceride, SFC, thermal behavior, and polycrystalline properties) were compared before and after CIE for the two different formulations. All margarine sheets were manufactured in an industrial production line to prove their industrial functionality and applications. The bending and sheeting characteristics of products were also evaluated to guide the subsequent industrial production of sheet margarine.

## 2. Materials and Methods

### 2.1. Materials

Shea olein (SHOL, Iodine Value (IV) 59–63 g I_2_/100 g oil) was purchased from the PGEO group in Malaysia, palm stearin (PSt, IV 33–35), palm kernel olein (PKOL, IV 22–24), palm olein (POL, IV56–58) were provided by Yihai Kerry in Shanghai, China. N-hexane and isopropanol were of chromatography HPLC grade. Other reagents were of analytical grade. Monoacylglycerol (MAG) and lecithin were purchased from Danisco (Kunshan China). Potassium sorbate was from Daicel Food Ingredients (Nanning, China) Co. Ltd., Sodium methylate(MeONa), citric acid (CA), both potassium hydroxide and methanol were analytical pure, and purchased from Shang hai Aladdin Reagent Co. Ltd. β-carotene was procured from BASF (Shanghai, China), and butter flavors from Givaudan (Shanghai, China) Co., Ltd.

### 2.2. Products Preparations

Formula NIEA was a conventional sheet margarine formula consisting of 50 wt%PSt, 15 wt% PKOL, and 35 wt% POL, while in formula NIEB, 35 wt% POL was replaced with 35 wt% SHOL. β -Carotene was used as a colorant, and MAG and lecithin were used as emulsifiers in the oil phase. Potassium sorbate and citric acid were dissolved in the aqueous phase as preservatives. CIEA and CIEB were correspondingly interesterification samples based on the blend of oil of formula NIEA and NIEB, respectively. All the formulations are shown in Table 1.

The preparation method of CIE fat was as follows: 5 tons of base oil was blended first, then dehydrated at 105 °C under vacuum (*p* < 100 Pa) for 30 min in a reaction tank, then 0.1% (calculated by the mass of mixed oil) sodium methoxide was added, and the reaction lasted 40 min under vacuum (*p* < 100 Pa) at 95 °C. The 0.2% citric acid (10%, *w/w*) was added concerning the mass of the catalyst to terminate the reaction and washed to neutrality. After dehydration at 105 °C for 1–1.5 h under vacuum (*p* < 100 Pa), 1.5% activated clay was added to bleach the oil by continuously stirring for 25 min, and then filtered at 90 °C. Finally, the deodorization was carried out at 240 °C for 2 h. The RBD CIE oils were the base oils for sheet margarine production.

All ingredients were blended according to the recipes presented in Table 1, After pasteurization, the emulsion was held at 55 ± 2 °C, ready for chilling.

Four ISMs (coded NIEA, CIEA, NIE B, and CIEB), in 3-ton batches, were produced in an industrial margarine line (Nexus 244 LC, SPX Technologies) equipped with four chilling tubes (called C_1_, C_2_, C_3,_ and C_4_), two pin working units (P_1_, P_2_), and one resting tube (RT). In the production, the pipe configuration was C_1_-C_2_-P_1_-C_3_-C_4_-RT, and only one pin worker was used. The flow rate was 2500 L/h and the rotational speed of scrapers and pin working unit was 490 rpm and 100 rpm, respectively. One pin working unit (Volume 100 L) was located between the second and third chilling units. The temperatures of the cooling medium (CO_2_) were set to −5, −5, −10, and −10 °C for the first, second, third, and fourth chilling units, respectively. All the products were tempered at 25 °C for 5 days and refrigerated at 5 °C for further evaluation. The detailed processing parameters are listed in Appendix A.

### 2.3. FA Composition

Fatty acid methyl ester (FAME) was prepared refers to AOCS CE 2-66 standard [27]. FAME analysis was performed by a Gas Chromatography (GC) system (Agilent CP-sil 88, 50 m × 0.25 mm × 0.2 µm, Santa Clara, CA, USA) with a flame ionization detector (FID) and chromatographic column according to the method reported by Gao et al. [17]. All determinations were performed three times and the means ± standard deviations were reported.

### 2.4. TAG Composition

TAG composition of different oils was analyzed by GC according to AOCS Official Methods (AOCS 1997). Three determinations were performed and the average was taken.

### 2.5. SFC

SFC was determined by following the AOCS Official Method Cd 16-81 using a pulsed nuclear magnetic resonance (pNMR) (Bruker MQ 20, Saarbrücken, Germany).

For fat blends, 3.0–4.0 g fat was melted and transferred into an NMR tube in triplicate. The samples were tempered at 80 °C for 30 min, followed by 0 °C for 60 min for complete crystallization of the oil sample. The tube was held for 30 min in the water bath with a pre-set temperature of 10, 20, 25, 30, 35, and 40 °C before measurement. Three measurements were performed, and the average values were reported.

For sheet margarine, first, about 3.0–4.0 g margarine was plunged into an NMR tube with a dedicated plunger. Placed the NMR tubes into 10 °C and 20 °C incubators for 24 h, then SFC was determined. Each sample was measured three times and the average value was taken.

### 2.6. Microstructure Analysis by Polarized Light Microscope (PLM)

A transparent sheet of sample on the glass slide was prepared by drawing an appropriate quantity of sample with a capillary tube and pressing on the glass slide, which was observed using a PLM microscope (Nikon Eclipse E 400, Tokyo, Japan) with the 50 × 10 magnification by following the method reported by Gao et al. [17], and the pictures were further processed by the image software.

### 2.7. Thermal Properties by DSC

The thermal behaviors of the samples were analyzed using differential scanning calorimetry (DSC, TA Q2000, New Castle, DE, USA) by following the method reported by Zhang et al. [9]. With nitrogen flowing at 50 mL/min, the program of test temperature was (a) isothermal holding for 5 min at 5 °C, then heating from 5 to 80 °C at 5 °C/min, (b) isothermal holding 80 °C for 5 min, then cooling from 80 °C to −20 °C at 5 °C/min, (c) isothermal holding −20 °C for 5 min, and then heating from −20 to 80 °C at 5 °C/min.

### 2.8. Crystallization Rate

The crystallization rate of each sample was investigated at different temperatures (5, 10, 15, and 20 °C) using a mini pNMR analyzer (Bruker MQ 20, Saarbrücken, Germany) by following the method reported by Zhang et al. [9]. The SFC was measured at intervals of 2 min for 40 min.

### 2.9. Polymorphism by X-ray Diffraction (XRD)

A Bruker D8 advance diffractometer (Bruker, Saarbrücken, Germany) with a copper X-ray tube (λ Cu = 1.54 Å) was used to study the polymorphism of sheet margarine operating at 40 kV and 40 mA. The scanned angle ranged from 12° to 30° and the scan speed was 2°/min with a step increment of 0.02° according to the method reported by Gao et al. [17].

The percentage of β type and β ‘type crystals in samples was calculated by normalization of peak area, and the peak background was removed during normalization. The content of β crystal was calculated from the area of short spacing peaks at 4.60 Å and β ‘crystal was calculated from the area of short spacing peaks at 4.20 Å and 3.80 Å.
%β = A_β_/(A_β_ + A_β ‘_) × 100, %β ‘ = A_β ‘_/(A_β_ + A_β ‘_) × 100(1)
where A_β_ and A_β ‘_ is the area of β type and β ‘type, respectively.

### 2.10. Hardness

The sheet margarine was stored at 10 °C, 15 °C, 20 °C, and 25 °C incubators for 1 day, and the hardness at the corresponding temperature was evaluated using a Texture Analyzer (TA. XT 2i/5, Stable Micro Systems, London, UK) with a P/6 probe by following the method reported by Gao et al. [17]. It was conducted at a penetration rate of 2 mm/s to a depth of 10 mm, and the maximum force value was recorded as the hardness. Each sample was measured three times, and the average value was taken as the final hardness value.

### 2.11. Bending Properties

Refrigerated sheet margarine was put into 10 °C, 15 °C, and 20 °C incubators overnight, and then bent at 90 degrees to observe the bending of the samples.

### 2.12. Sheeting Properties

The refrigerated sheet margarine was put into 10 °C, 15 °C, and 20 °C incubators overnight, and was compressed using the dough sheeting machine. The thickness of sheet margarine was sheeted down from 20 mm → 15 mm → 10 mm → 6 mm → 4 mm, to observe the crispness of the sample by following the method reported by Gao et al. [17].

### 2.13. Statistical Analysis

Statistical analysis was conducted by one-way ANOVA with the SPSS software. Variation of the significance level is 0.05 (*p*-value) according to Duncan’s Multiple Range Test.

## 3. Results and Discussions

### 3.1. Fatty Acid Composition (FAC)

As shown in Table 2, NIEA and CIEA were rich in palmitic acid (C16:0, 45.71%) and oleic acid (C18:1, 33.73%), while NIEB and CIEB were the same as the former, but contained more stearic acid (C18:0, 14.67%) with 10% less palmitic acid, which was the main difference in FACs. The minute amounts of trans fatty acids derived from deodorization rather than chemical interesterification were observed [28]. Interesterification can alter the arrangement of acyl groups without producing TFAs.

### 3.2. TAG Composition

The TAGs of the products before and after CIE are changed significantly as shown in Table 3. Generally, the properties of margarine depended on the appropriate triacylglycerol composition [6] which is necessary to monitor due to random interesterification. SFC, slip melting point and hardness are closely related to the TAG composition [29]. Several tri-saturated TAGs (PPP, MLP, and PLP) are slightly altered by CIE but the total content did not modify significantly in both formulations. Interestingly, the PPP content in formula NIEB decreased compared to formula NIEA. The reason behind the difference in the palmitic acid content (45.71% and 34.03% respectively) in both formulations is the presence of SHOL from formula B rather than palm olein. The literature data revealed that SHOL contained a higher content of MUFA and SU_2_ TAGs and enabled it to be a potentially healthier liquid oil in the specialty fats field [9]. In addition, the reduction in U_3_ content (from 7.33% to 2.13%) in formula NIEB may increase SFC. The content of S_2_U and SU_2_ was reduced in formula A while a slight increase in formula B can be observed. Although the total amount of S_2_U and SU_2_ did not change much, the exchange of fatty acids has altered the properties of margarine greatly.

### 3.3. SFC

The SFC of products is an important basis to evaluate characteristics of the product, including appearance texture (greasiness and homogeneity), and physical properties (malleability and fusibility) of solid fat products [30], and it is mostly influenced by the triacylglycerol composition of the fats. As described in Table 3, the formulations with SHOL instead of palm olein possessed a higher level of unsaturated fatty acids and led to a lower SFC at each respective temperature. The transesterification resulted in a higher and lower value of SFC of the fat base at a lower (below 25 °C) and higher (above 30 °C) temperature, respectively as shown in Figure 1a, indicating the occurrence of the acyl exchange among the different melting point triacylglycerols in the transesterification process. These outcomes revealed that the CIE fat presented a steeper SFC curve as reflected in the DSC melting curve hereinafter, presenting consistency with the previous reports of Samuel and Joy [31]. The SFC values of the finished product are shown in Figure 1b. The SFC of the finished product is relatively stable although the value is slightly diminished upon the addition of water, emulsifiers, and colorant. The results are consistent with previous studies to indicate that the SFC range from10 to 40% over a range of 33.3–10.0 °C can enable better laminating ability and layering effect during production in the baker [6,29,32]. Moreover, the SFC of the product developed from formula NIEB/CIEB with SHOL addition is lower either at 10 °C or 20 °C than NIEA/CIEA, indicating a significant reduction in the SFC value of the products with the addition of SHOL. The SFC values of final products manufactured with the formula CIEA are all higher than formula NIEA, which is in line with the trend of the oil-based SFC curve. However, the SFC value of CIEB at 10 °C is less than NIEB, presenting no consistency with the oil base. The possible reason behind this phenomenon may be the impact of additives on lipid crystallization. Further explorations are also necessary to understand the relationship between additives and lipid crystallization for a better understanding of functionality.

### 3.4. Microstructure (PLM)

The different TAG composition and polycrystalline form of triglyceride would remarkably affect its microstructure, thus influencing the macroscopic characteristics of the product which are determined by the microstructure of the crystal [33]. Figure 2 showed the microstructure of crystals in different samples at 500× magnification. Comparing NIEA/NIEB with CIEA/CIEB, the number of crystals increased significantly after transesterification which is corroborated by the SFC value. The polyunsaturated triacylglycerols transformed into monounsaturated triacylglycerols after CIE and later crystallized easily at low temperatures, as shown in Table 3 and Figure 2. The result explained the larger quantity of crystallization of sample NIEB than that of sample A even having the lower SFC value of sample B, implying the stronger crystal network of B/CIEB to improve the bending and malleability of the product. Interestingly, the PLM image of CIEA showed a large aggregation of crystals, which may be its tendency to form granular crystals [3]. After CIE, smaller crystals are obtained to form a more viscoelastic network of fat crystals, making the product glossier and more resilient.

### 3.5. Thermal Properties by DSC

The changes in triacylglycerol composition can be indirectly reflected by differential scanning calorimetry (DSC). Figure 3a showed the melting curve of the products. The melting peaks of medium and high melting point components of several samples are difficult to separate, and a broad peak appeared at 20–40 °C, indicating the complex composition of triacylglycerols in several formulations is complex and the existence of multiple crystal nuclei. After transesterification, the small peaks of components with high melting points in NIEA/NIEB disappeared, and the melting temperature decreased, indicating the reduction in the high melting points glycerides (S_3_). Figure 3b showed the crystallization curves of products with different formulations from 80 °C to −20 °C. Two crystallization peaks are observed in NIEA and NIEB, which reflected the composition of triacylglycerols with different melting points in the oil base, respectively. There is a small peak in the crystallization curve which may be derived from the intermediate melting point component formed after CIE, which infers more complexity in the component after transesterification. The initial crystallization temperature of NIEB/CIEB is higher than that of NIEA/CIEA due to the addition of SHOL. For the high melting point components, the crystallization temperatures of the samples after CIE are lower than that of the samples without transesterification, although the peak temperature of the samples is almost the same. However, it exhibited the opposite results for the low melting point components, following similar phenomena as reported by Oliveira and Rodrigues [18]. Simultaneously, sharp peaks of the components with low melting points are observed in NIEA/NIEB samples without transesterification as shown in Figure 3c, summarizing the relatively simple composition of the TAG at this stage. After going through transesterification, the sharp peaks of the melting curve of the finished product disappeared due to the complex composition derived from the medium melting point components.

### 3.6. Crystallization Rate

The characteristics of nucleation and growth of crystals are revealed in the isothermal crystallization curves of the samples. Figure 4 showed the crystallization behaviors of samples before and after transesterification at different temperatures (5 °C, 10 °C, 15 °C, and 20 °C, respectively). Before transesterification, Sample NIEA presented an S-shaped curve at all temperatures, rapidly crystallized at the beginning, and subsequently slowed down. After a short plateau, the crystallization rate increased again and finally reached the crystallization equilibrium state. Previous studies speculated that this phenomenon was caused by the separate crystallization of triglycerides with different melting points [34] and proven by the DSC crystallization curve which presented a big gap between the low-temperature peak and high-temperature peak. After transesterification, CIEA crystallized more rapidly at all investigated temperatures and reached a plateau preferentially compared with other samples.

Similarly, at all investigated isothermal crystallization temperatures, the initial crystallization rate of CIEB was slower than sample NIEB within the first 2 min, after which the crystallization rate exceeded and finally reached the crystallization equilibrium point with a higher SFC value. The crystallization rate of sample B is relatively slower compared with that of sample A due to the addition of SHOL led to lower contents of S2U and SU2 and higher content of U3 in sample NIEB, although both have similar contents of saturated fatty acids (Table 2).

All transesterified samples reached the crystallization equilibrium more quickly with a higher SFC value, which is consistent with the SFC results in Figure 1. The results demonstrated that the crystallization rate of triacylglycerol is accelerated after transesterification, which is more favorable to crystal growth and the quick formation of crystal network in ISM production.

### 3.7. Crystal Morphology

The polymorphism is identified using an X-ray diffraction pattern. The crystal forms are monitored by the short spacing as shown in Figure 5. The peak appearing at 4.15 Å is the α-form, the peak at 4.2 Å and/or 3.8 Å is the β′-form and the peak at 4.6 Å is the β-form. β′-form is the most desired crystal for pastry fat [35]. Sheet margarine with preponderant β′-form exhibited a more homogeneous texture and better malleability owing to the larger specific surface of the smaller and uniform β′-form, enabling it to bind a larger number of liquid oils and decreased the phase separation. While the β-form crystal is relatively coarse and aggregated and the crystal size tended to grow with time, resulting in the poor ability to immobilize liquid oil. Polymorphism is greatly influenced by thermal treatment. More β’ crystals are usually obtained by fast cooling compared with slow cooling. Therefore, the strength of chilling is one of the most important parameters in the manufacture of ISM.

Table 4 showed the short-space X-ray diffraction pattern of products before and after transesterification. In sample NIEA, the triglycerides are mainly a mixture of β polycrystalline (29.14%) and β′ polycrystalline (70.86%). After transesterification (CIEA), only one peak appeared between 4.2 Å and 4.4 Å indicating the reduction and increment in the proportion of β polycrystalline (13.55%) and β′ polycrystalline (86.45%), respectively to contribute the better malleability. Meanwhile, a predominant β’ form existed and a rare transformation (from β′ to β) is observed in both NIEB and CIEB, with β′ percentage of 92.29% and 93.72%, respectively, which may be due to the addition of SHOL. Meanwhile according to the DSC melting curve, compared with CIE products, the presence of melting peaks at high temperatures indicates that there are more β crystals in NIE products, and it can also be inferred that β crystal types decrease after CIE. After transesterification, the two peaks at 4.2 Å and 4.4 Å are replaced by a larger peak, and more β′ crystal was formed which is confirmed by corroborating with the PLM diagram and SFC curve which can improve the physical properties of sheet margarine.

### 3.8. Hardness

Hardness is closely related to temperature variation. Therefore, all temperatures should be strictly controlled throughout ISM production. The composition of TAG, SFC value, and polymorphism also have significant impacts on the hardness of the end product. The malleability and plasticity of sheet margarine can be preliminarily assessed by the hardness. The malleability and plasticity characteristics are highly relied on in the product’s nature such as being too hard and soft can result in brittle surface and oil leakage, respectively. The hardness value measured by TA is shown in Figure 6a. All the samples were stored at 10 °C, 15 °C, 20 °C, and 25 °C for one day prior to measurement. The SFC value of CIEA was higher than that of NIEA but the hardness displayed an opposite trend at a temperature below 20 °C which may be due to the major role of polymorphism in hardness and sample NIEA without CIE contained more β form. However, the SFC played a significant role in hardness with the increment in the temperature. At 25 °C, the hardness of CIEA (285.401 g) is higher than NIEA (177.019 g) due to its higher SFC value. Similarly, the hardness of sample NIEB at low temperatures (5 °C, and 10 °C) is higher but lower at 20 °C and 25 °C than CIEB. The aforementioned results explained a wider temperature tolerance of the product with CIE.

Figure 6b showed the variation of hardness with a function of time at 20 °C after the samples were taken out from 10 °C. As mentioned above, CIE products CIEA/CIEB processed lower hardness than NIEA/NIEB. The NIEA/NIEB had a tremendously high hardness value at 10 °C of about 3137 g and 2060 g, respectively which remarkably decreased to 1074 g and 241 g, and the reductions are up to 66%, and 88%, respectively within 2 h. Meanwhile, the hardness of CIEA/CIEB decreased from 1480 g and 863 g to 642 g and 234 g (56%, 72% reduction), respectively. CIE products have a wider plastic range than non-CIE products and are more conducive to the actual industrial production of bakery products.

### 3.9. Bending Properties

To further evaluate the malleability and brittleness of products, the sheet margarine was folded inward 90 degrees and its bending characteristics were observed. Samples with higher hardness were more easily broken. As shown in Figure 7, The bending capacity of the product increased significantly as the temperature increased from 10 °C to 20 °C due to the decrease in hardness and all of the samples did not fracture at ≥20 °C. Compared to sample NIEA, which was broken at 15 °C, CIEA did not break despite a few cracks appearing on the surface. The results revealed that the transesterification process can improve the bending capacity of the product. Sample NIEB showed a similar fracture compared with sample NIEA at 10 °C although it had lower hardness and β crystal content than sample NIEA. However, its bending property improved compared to that of NIEA, even presenting better than CIEA with the increase in temperature. The CIEB exhibited the best bending ability which may be linked to the addition of SHOL with lower hardness. CIEB showed no fracture at 10 °C and the bending is smooth at 15 °C, proving better plasticity and was more conducive to actual industrial processing.

### 3.10. Malleability of Product

The malleability of the products stored at different temperatures was tested using a sheeter to gradually laminate the fat to the thickness of 4 mm, and then observed the breakage of the fat belt. Owing to its high hardness and β-form crystal, product NIEA has very poor malleability at low temperatures. As shown in Figure 8, a large area of fracture occurs at 10 °C, many small notches appeared at 15 °C and slight notches were still present even at 20 °C although having a smooth surface during the laminating process. In general, the malleability of all samples gradually improved with the increase in temperature which is similar to the bending results. However, CIEA exhibited good malleability at low temperatures probably due to the decrease in saturated fatty acid content after transesterification, thus leading to lower hardness. 

The TAG with higher saturated fatty acids is more likely to form β crystals due to the formation of initial crystal nuclei by saturated TAG, which further promoted the continuous crystallization of POP/SOS and other TAGs [32]. Sample NIEB showed better physical properties than sample NIEA, this may be in connection with the replacement of palm olein with SHOL leading to the presence of fewer β-form crystals. Sample NIEB can be laminated to thin slices at 10 °C without breakage although the demonstration of edges with uneven thickness and some defects. A smooth and continuous fat belt can be achieved when sheeted at 15 °C and 20 °C. After transesterification, the CIEB showed good malleability at all temperatures, which is more suitable for food industry production.

## 4. Conclusions

This study demonstrates the difference in physicochemical properties and application functionalities of ISM manufactured with two formulations and their interesterifications. Four ISMs are evaluated on the modification of FAC and TAG, polymorphism, thermal behavior, and malleability. The results show that sheet margarine with the addition of SHOL to replace palm olein has more ideal polymorphism, softer texture, and better malleability. The physical properties of the sheet margarine can be further improved by the CIE process. The lower hardness of CIEA/CIEB compared with NIEA/NIEB at the corresponding temperatures contribute to better bending ability and is less likely to fracture due to the increasing ratio of β′ crystals resulting from the change in the composition of triglyceride although the SFC value increased slightly. Moreover, CIEA and CIEB samples have faster crystallization rates and can quickly form homogeneous and strong crystal networks to support the structure of sheet margarine. DSC curves of each sample after transesterification show that the crystallization proportion from the medium melting point component increases while slightly decreasing from the high and low melting point components resulting from the modification of TAG. At the same time, an increase in the crystallization, and reduction in the crystal size after CIE is observed according to the SFC curve and PLM diagram. Due to more β‘ crystals and lower hardness, the transesterification samples showed better bending characteristics and malleability in industrial production, which is more desired in commercial bakery production. Therefore, formulation with SHOL and interesterification process has advantages in improving the malleability of the product and can be employed as an effective method to manufacture industrial sheet margarine with zero trans fatty acids.

## Figures and Tables

**Figure 1 foods-11-03592-f001:**
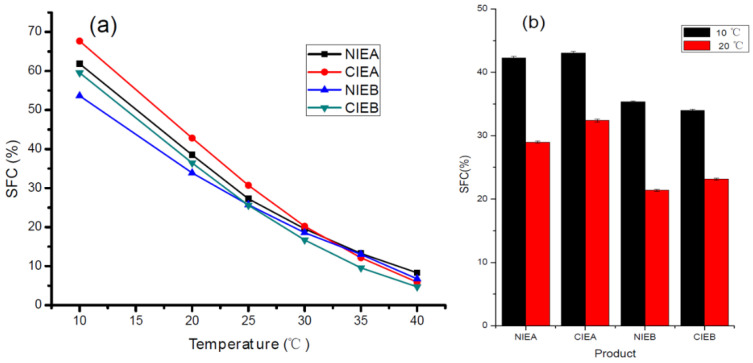
SFC of four fat blends (**a**) and the SFC of four final sheet margarine at 10 and 20 °C (**b**).

**Figure 2 foods-11-03592-f002:**
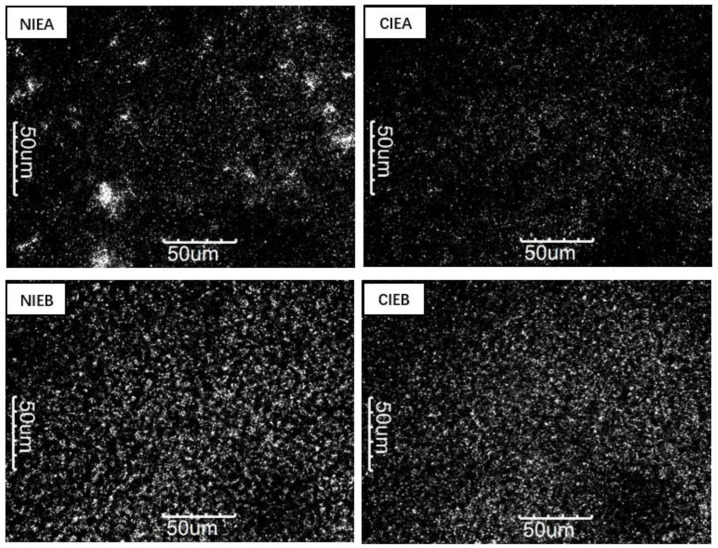
PLM images of NIEA, CIEA, NIEB and CIEB.

**Figure 3 foods-11-03592-f003:**
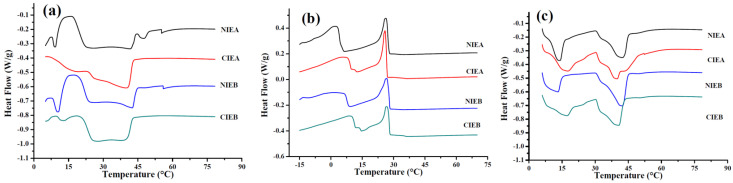
DSC curve of four ISMs, melting curve (**a**) and crystallization curve (**b**), re-melting curve (**c**).

**Figure 4 foods-11-03592-f004:**
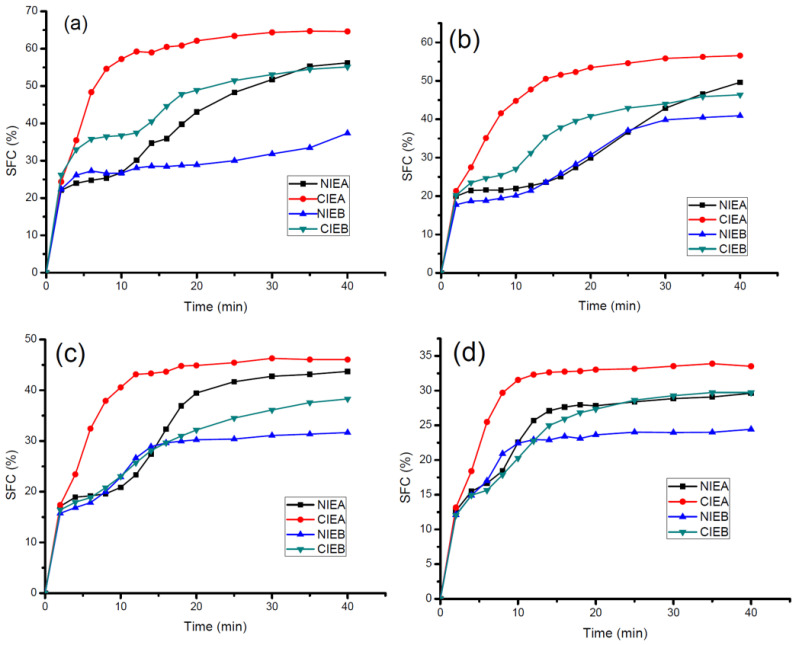
Isothermal Crystallization rate of sheet margarine at 5 °C (**a**), 10 °C (**b**), 15 °C (**c**), and 20 °C (**d**).

**Figure 5 foods-11-03592-f005:**
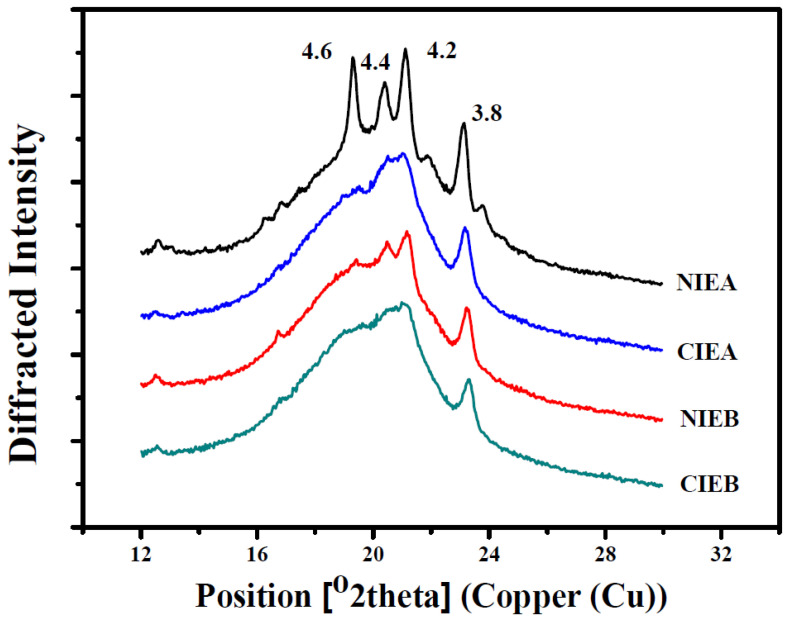
XRD spectra of samples before and after interesterification.

**Figure 6 foods-11-03592-f006:**
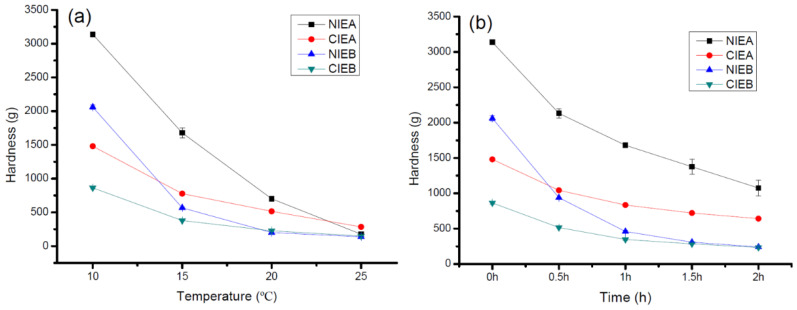
The hardness of samples at different temperatures (**a**) and the hardness changes vs. time (**b**) when refrigerated samples were placed at 20 °C.

**Figure 7 foods-11-03592-f007:**
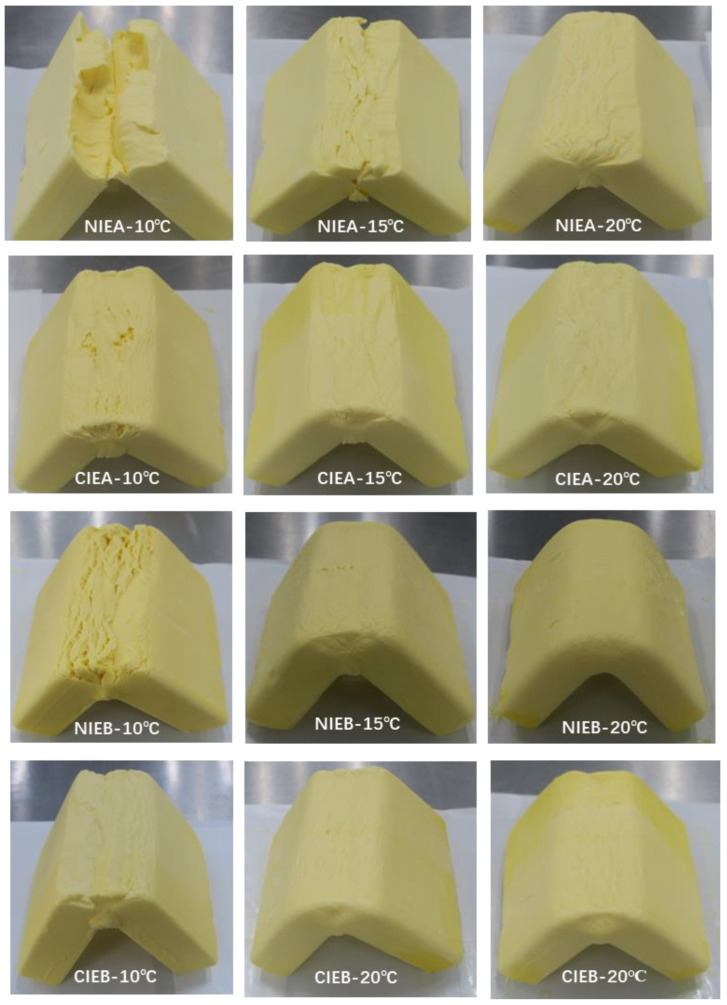
Bending abilities of samples at 10 °C, 15 °C, and 20 °C.

**Figure 8 foods-11-03592-f008:**
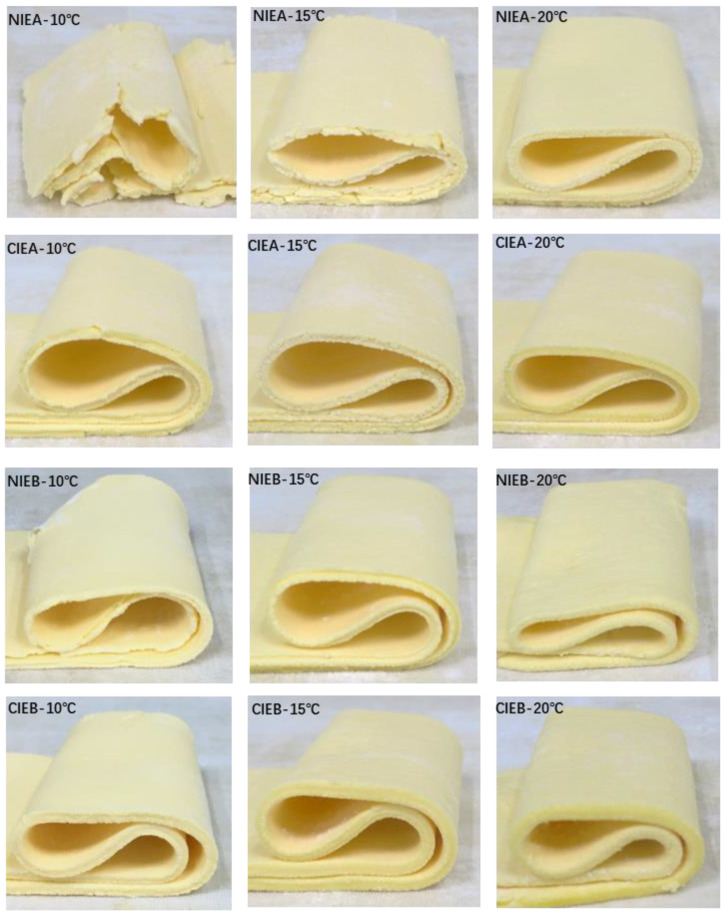
Malleability of different samples at 10 °C, 15 °C, and 20 °C.

**Table 1 foods-11-03592-t001:** Commercial Formulations of samples NIEA, NIEB, CIEA, and CIEB.

Ingredients (%)	NIE *A	NIEB	CIEA	CIEB
Palm Stearin (PSt)	40.40	40.40		
Palm Kernel olein (PKOL)	12.12	12.12		
Palm olein (POL)	28.28			
Shea olein (SHOL)		28.28		
CIE (50PSt+15PKOL+35POL)			80.80	
CIE (50PSt+15PKOL+35SHOL)				80.80
Subtotal fat blends	80.8	80.8	80.8	80.80
Monoacylglyceride (MAG)	1.00	1.00	1.00	1.00
Lecithin	0.50	0.50	0.50	0.50
β-carotene	0.00	0.00	0.00	0.00
Butter flavor	0.10	0.10	0.10	0.10
Water	16.00	16.00	16.00	16.00
Salt	1.50	1.50	1.50	1.50
Potassium sorbate	0.09	0.09	0.09	0.09
Subtotal additives	19.20	19.20	19.20	19.20
Total	100	100	100	100

* Note: NIE = Not interesterified.

**Table 2 foods-11-03592-t002:** FACs of two different formulas before and after CIE (wt.%).

	Formula A(NIEA/CIEA)	Formula B (NIEB/CIEB)
C8:0 *	0.53 ± 0.04	0.49 ± 0.03
C10:0	0.56 ± 0.02	0.61 ± 0.01
C12:0	5.11 ± 0.33	5.77 ± 0.27
C14:0	2.78 ± 0.15	2.56 ± 0.03
C16:0	45.71 ± 1.24	34.03 ± 1.74
C18:0	4.84 ± 0.24	14.67 ± 0.79
C18:1t	0.39 ± 0.01	0.25 ± 0.01
C18:1	33.73 ± 1.34	35.75 ± 1.08
C18:2t	0.16 ± 0.01	0.22 ± 0.01
C18:2	7.26 ± 0.14	6.04 ± 0.21
C20:0	0.20 ± 0.01	0.49 ± 0.01
C18:3	0.26 ± 0.02	0.29 ± 0.01
ΣTFA	0.55 ± 0.01	0.47 ± 0.02
ΣSAFA	58.62 ± 2.13	57.51 ± 0.04
ΣMUFA *	34.12 ± 1.31	36.00 ± 2.17
ΣPUFA *	7.68 ± 0.15	6.55 ± 0.22

* Abbreviations: C8:0, octanoic acid; C10:0, decanoic acid; C12:0, lauric acid; C14:0, myristic acid; C16:0, palmitic acid; C18:0, stearic acid; C18:1, oleic acid; C18:1t, trans-oleic acid; C18:2, linoleic acid; C18:2t, trans-linoleic acid; C18:3, linolenic acid; C20:0, arachidic acid; MUFA: Mono-unsaturated fatty acid; PUFA: polyunsaturated fatty acid. Note: Values are the means ± standard deviations.

**Table 3 foods-11-03592-t003:** TAG composition(wt.%) of two different formulations before and after CIE.

TAGs	NIEA	CIEA	NIEB	CIEB
LaLaLa	4.46 ± 0.14	1.41 ± 0.07	4.94 ± 0.33	1.54 ± 0.07
LaLaM	4.01 ± 0.13	2.28 ± 0.12	3.93 ± 0.23	3.23 ± 0.12
LnLO/CPO	1.70 ± 0.03	3.87 ± 0.15	1.96 ± 0.03	3.82 ± 0.17
MMO/LLP	1.16 ± 0.01	8.26 ± 0.24	1.44 ± 0.01	6.52 ± 0.23
MPP	1.12 ± 0.01	3.25 ± 0.14	1.17 ± 0.01	3.83 ± 0.05
MOM	0.91 ± 0.01	7.70 ± 0.03	1.01 ± 0.01	6.84 ± 0.11
PPP	15.30 ± 0.24	10.01 ± 0.25	13.27 ± 0.34	5.57 ± 0.12
MOP	1.66 ± 0.02	3.63 ± 0.02	1.30 ± 0.01	4.52 ± 0.11
MLP	0.85 ± 0.03	2.96 ± 0.01	1.79 ± 0.01	4.10 ± 0.12
PPS	0.85 ± 0.01	2.53 ± 0.01	1.65 ± 0.01	6.53 ± 0.11
POP	31.01 ± 1.05	19.39 ± 0.78	24.56 ± 0.93	12.30 ± 0.56
MOO	0.72 ± 0.02	0.91 ± 0.01	0.47 ± 0.01	1.06 ± 0.01
PLP	6.75 ± 0.09	4.06 ± 0.17	5.70 ± 1.34	2.19 ± 0.01
PSS	0.22 ± 0.01	2.36 ± 0.01	0.43 ± 0.01	4.41 ± 0.02
POS	4.65 ± 0.13	5.97 ± 0.01	4.36 ± 0.25	9.43 ± 0.33
POO	12.66 ± 0.54	10.14 ± 0.37	9.33 ± 0.32	8.95 ± 0.27
PLS	1.22 ± 0.01	0.94 ± 0.01	1.19 ± 0.01	1.46 ± 0.02
PLO	4.97 ± 0.02	4.38 ± 0.03	2.68 ± 0.27	2.94 ± 0.18
SOS	1.26 ± 0.01	0.25 ± 0.01	0.29 ± 0.01	2.35 ± 0.01
SOO	--	0.79 ± 0.01	3.34 ± 0.31	2.91 ± 0.02
OOO	1.66 ± 0.01	1.65 ± 0.03	7.33 ± 0.74	2.13 ± 0.03
SLO	0.56 ± 0.01	0.41 ± 0.01	2.66 ± 0.24	0.98 ± 0.01
OLO	0.82 ± 0.02	1.06 ± 0.02	1.62 ± 0.01	1.05 ± 0.01
OLL	0.18 ± 0.01	0.21 ± 0.01	1.11 ± 0.01	0.13 ± 0.01
Others	0.25 ± 0.01	0.61 ± 0.01	0.18 ± 0.01	0.22 ± 0.01
ΣU_3_	1.67 ± 0.05	1.65 ± 0.07	7.33 ± 0.18	2.132 ± 0.14
ΣSU_2_	14.20 ± 0.31	12.89 ± 0.42	12.97 ± 0.48	14.05 ± 0.55
ΣS_2_U	43.36 ± 2.04	39.31 ± 1.85	34.34 ± 1.88	36.96 ± 1.79
ΣS_3_	31.77 ± 1.56	27.38 ± 1.33	32.53 ± 1.57	28.38 ± 0.97

Abbreviation used: La lauric acid; M, myristic acid; P, palmitic acid; L, linoleic acid; Ln, linolenic acid; O, oleic acid; S, stearic acid; A, arachidic acid; U_3_, tri-unsaturated TAG; SU_2_, di-unsaturated TAG; S_2_U, Monounsaturated TAG; S_3_, tri-saturated TAG.

**Table 4 foods-11-03592-t004:** Polymorphic forms (%) of crystals of products.

%	NIEA	CIEA	NIEB	CIEB
β′	70.86 ± 0.57	86.45 ± 0.78	92.29 ± 0.83	93.72 ± 0.91
β	29.14 ± 0.16	13.55 ± 0.11	7.71 ± 0.12	6.28 ± 0.12

## Data Availability

Data is contained within the article or Appendix A.

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
