# Peer review of "Malleability and Physicochemical Properties of Industrial Sheet Margarine with Shea Olein after Interesterification"

_foods, 2022, doi:10.3390/foods11223592_

Round 1

Reviewer 1 Report

The article under review is devoted to an actual topic - to improve the malleability of Industrial Sheet Margarine (ISM) in industrial level production with shea Olein (SHOL), rich in stearic acid C18:0 and the chemical interesterification (CIE) fat.

The manuscript has been well written (coherence, flow, and readability). The materials of the study are appropriately selected and performed.

The title and abstract are appropriate for the content of the text.

The authors have a good work, I enjoyed reading it.

This is an interesting study, both in the introduction and methodology (a section that presents a substantial amount of information) and in the discussions related to the results (with a precise and extended presentation).

The paper is well documented, is a hallmark of a scientific research team and also demonstrates the importance of methodological knowledge in research studies.

Overall the results are compelling, and the study showed that formulation with SHOL and interesterification process has advantages in improving the malleability of product and can be employed as an effective method to manufacture industrial sheet margarine with zero trans fatty acids.

Reviewer 2 Report

The research was well designed and conducted.

The research is of interest to the area of oils and fats, finding results of interest in what is expected. The methodology was adequate to the research.

It is requested to identify the samples in all figures and tables, for better understanding of the reader.

It is requested to comment on the cost of the type of fat suggested and studied, comparing it with market fats, to obtain the commercial viability of the suggested products.

Reviewer 3 Report

The manuscript presents an interesting study on the use of shea oil for the production of sheet margarine. The work was well structured and used several advanced analytical tools to support the conclusions. The text is clear and easy to read. However, two important aspects must be discussed by the authors. First, one of the reasons for using shea oil in the formulation is to reduce the content of trans fatty acids. However, the trans levels shown in Table 2 are very high and are not in accordance with most of the legislation in force in several countries (for example, the European Union).  The simple removal of these trans acids can change the properties of the studied fats. In what process were trans acids formed? Would it be possible to remake the product without having such high levels of this undesirable component? Subsequently, the authors consider the beta, beta prime, and alpha forms as inherent to the oil composition. However, they are closely linked to the temperature variation to which the fat was subjected. The authors should redo the discussion with this in mind. Finally, find attached a file with minor comments about the text. Considering the above, I recommend the manuscript for publication after minor revisions.

Reviewer 4 Report

Malleability and physicochemical properties of Industrial Sheet Margarine with Shea Olein after Interesterification

Comments:

The article is well-structured and well-written.

Following are some observations:

Recent work related to this topic should be added.

Line 67: is the main means adopted to prepare zero TFA??? What is meant by main means?? Rephrase it.

Line 118: 95 °C?? There should be no space between value and degree sign.

Line 120 and 122: There should be no space between value and degree sign.

Line 132: CO2?? It should be CO2

Line 134: The detailed Processing Parameters?? Why in capital letters? It should be processing parameters.

Line 138: GC? Write full form when using for the first time.

Line 167: remove double space

Line 169: Thermal properties (DSC)???  How can you abbreviate Thermal Properties with DSC? It is the name of the instrument by which you have measured. Please correct it.

Line 149-Line 211: you have not mentioned any reference which you have followed for the tests. Do mention the references for each test you have performed? For example, you have measured SFC by nuclear magnetic resonance by following which method??

In whole manuscript, you have given space while mentioning temperature like, 25 ℃. Remove this space. It should be 25℃.

Too many mistakes in the references!

Somewhere you have used the full form of journal names and somewhere you have used an abbreviated form. It should be in a single format.

Why you are using et al in the references?? Author names should be complete.

In some references, you have used journal names in abbreviated form. But the abbreviated format is not correct. For example, in 2nd reference, you have written Food Res Int, 2017 it is wrong. It should be Food Res. Int, 2017.

Kindly go through all the references, there are a lot of mistakes.

Round 2

Reviewer 4 Report

no more comments